# Intracellular *Staphylococcus aureus* Elicits the Production of Host Very Long-Chain Saturated Fatty Acids with Antimicrobial Activity

**DOI:** 10.3390/metabo9070148

**Published:** 2019-07-20

**Authors:** Natalia Bravo-Santano, James K. Ellis, Yolanda Calle, Hector C. Keun, Volker Behrends, Michal Letek

**Affiliations:** 1Health Sciences Research Centre, University of Roehampton, London SW15 4JD, UK; 2Division of Cancer, Department of Surgery and Cancer, Faculty of Medicine, Imperial College London, London W12 0HS, UK

**Keywords:** intracellular infection, lipids metabolism, fatty acids, antibacterial effect, *Staphylococcus aureus*

## Abstract

As a facultative intracellular pathogen, *Staphylococcus aureus* is able to invade and proliferate within many types of mammalian cells. Intracellular bacterial replication relies on host nutrient supplies and, therefore, cell metabolism is closely bound to intracellular infection. Here, we investigated how *S. aureus* invasion affects the host membrane-bound fatty acids. We quantified the relative levels of fatty acids and their labelling pattern after intracellular infection by gas chromatography-mass spectrometry (GC-MS). Interestingly, we observed that the levels of three host fatty acids—docosanoic, eicosanoic and palmitic acids—were significantly increased in response to intracellular *S. aureus* infection. Accordingly, labelling carbon distribution was also affected in infected cells, in comparison to the uninfected control. In addition, treatment of HeLa cells with these three fatty acids showed a cytoprotective role by directly reducing *S. aureus* growth.

## 1. Introduction

The Gram-positive *Staphylococcus aureus* is an opportunistic pathogen, known for its ability to cause nosocomial infections and infect immune-compromised people, e.g., cystic fibrosis patients. It is a facultative intracellular pathogen, carried by about one third of the global human population on the skin and/or in the nasal passages [1,2]. *S. aureus* is able to invade and proliferate within a wide range of mammalian cells, including both phagocytic and non-phagocytic cells [3,4,5,6]. During host cell invasion, *S. aureus* is able to induce profound rearrangements of the host cell cytoskeleton to favor its own internalization, control autophagy to facilitate its replication inside of a pathogen-containing vacuole, and manipulate apoptosis [6]. The intracellular mode of growth could be used by the pathogen to evade host immune recognition and avoid exposure to last-resort antibiotics, such as daptomycin, vancomycin and linezolid [7]. A better understanding of the host-pathogen interaction during intracellular infection could lead to novel therapeutic approaches that could be used in combination with traditional antibiotherapy.

Bacterial pathogens require accessible forms of chemical energy to support intracellular proliferation. Therefore, the interaction between intracellular pathogens and host cell metabolism is an important feature of virulence [8,9]. Bacteria need to adapt to the host microenvironment and, consequently, they exploit host metabolic pathways in order to ensure their survival and proliferation within host cells [10,11,12]. Most of the research on this topic has focused on well-studied intracellular pathogens such as *Shigella flexneri*, *Escherichia coli*, *Listeria monocytogenes*, *Salmonella typhimurium*, *Legionella pneumophila* and *Chlamydia trachomatis* [10,13]. For instance, *L. pneumophila* interacts with the host mitochondria, inducing a Warburg-like effect in the host cell to favor its own replication [14]. *S. flexneri* re-routes host central carbon metabolism through the glycolytic pathway to obtain an abundant nutrient-flux that allows its intracellular survival [15]. Moreover, intracellular *M. tuberculosis* requires host cholesterol import to persist inside both macrophages and mice’s lungs [16].

In contrast, very little is known about how the host cell metabolism is modulated and exploited by the pathogen *S. aureus*. The host metabolic response of human airway epithelial cells to *S. aureus* infection was recently investigated [17]. Interestingly, a reduction in nutrient uptake as well as an induction of nucleotide biosynthesis were observed in *S. aureus*-infected cells, although the underlying mechanisms are still unclear [17]. In addition, we recently found that intracellular methicillin-resistant *Staphylococcus aureus* (MRSA) reroutes host central carbon metabolism, leading to a starvation-induced autophagic flux in MRSA-infected cells [12]. Moreover, MRSA exploits the host AMPK-pathway during intracellular infection and pharmacological AMPK-inhibition hampered *S. aureus* intracellular proliferation.

Previously, fatty acids have been shown to act as powerful signaling molecules and be involved in different metabolic processes [18] that could be important for intracellular MRSA infection. Furthermore, antimicrobial activities have been described for lipids and particular fatty acids [19,20,21]. Additionally, host-derived fatty acids and lipids have an importance for intracellular infection as immune modulators [22]. On the other hand, *S. aureus* can incorporate host derived fatty acids into its own membranes [23]. To shed some light on the metabolism of lipids during cell infection, here we determined the levels and labelling carbon distribution of esterified fatty acids in HeLa cells after intracellular MRSA infection.

## 2. Results and Discussion

### 2.1. Levels and Labelling Pattern of Saturated Fatty Acids in HeLa Cells in Response to Intracellular MRSA Infection

We quantified nine lipid-bound fatty acids in our samples: arachidonic, eicosanoic, docosanoic, linoleic, myristic, oleic, palmitic, palmitoleic and stearic acids. Six of them showed no differences among treatments, such as stearic acid (Figure 1 and Appendix A). However, the absolute levels of two long-chain saturated lipid-bound fatty acids, docosanoic (C22:0) and eicosanoic acid (C20:0), as well as palmitic acid (C16:0), were markedly increased after intracellular MRSA infection (Figure 1). Levels of docosanoic and palmitic acid remained stable in cells exposed to heat-killed bacteria (Figure 1). In contrast, levels of eicosanoic acid significantly increased in cells exposed to heat-killed USA300 bacteria, although the increase was notably higher in MRSA-infected cells (Figure 1). This is suggesting that part of this increase is due to a host-immune-like response. Considering that mostly polyunsaturated eicosanoids have been related to inflammation and are considered a main component of the host innate immune responses [22], the rise of eicosanoic acid (a saturated fatty acid) in the host cell in response to heat-killed bacteria exposure is surprising.

Additionally, we performed carbon-labelling experiments in which one of the two major carbon sources in the medium (glucose and glutamine, respectively) was substituted by universally ^13^C-labeled varieties. This allows quantification of rate of synthesis and carbon flux. Interestingly, the label distribution varies across the three experimental conditions, especially for docosanoic and eicosanoic acids. In MRSA-infected cells, both eicosanoic and docosanoic acids showed a similar labelling distribution, displaying an increase in glucose-derived label (Figure 2 and Appendix A). The majority of label was detected in the *m* + 2 and *m* + 4 isotopologues (Appendix A). ISA flux analysis [24] indicated that these long-chain fatty acids were mostly the result of fatty acid elongation of existing fatty acids rather than de novo synthesis of whole molecule (model parameters for eicosanoic acid from infection HeLa cells grown in ^13^C_6_ glucose-labelled cultures—fractional enrichment *D* 0.44; de novo synthesis, *g*(*t*), 0.26; elongation 0.74). This is broadly in agreement with the fact that we see no or little increased enrichment in palmitate or stearate. As ISA flux is ideal for modeling the production of saturated long and very-long chain fatty acids, we also used the recently developed FASA package [25]. The resulting parameters are broadly comparable, though they show a lesser dependence on elongation (*D* 0.6; synthesis 0.31; import (present before addition of label) 0.34; elongation (combined) 0.35). The increase in glucose-derived label of both eicosanoic and docosanoic acids agrees with the fact that elongation of fatty acids is driven from glucose [26]. In accordance with this observation, our previous study on the metabolism of HeLa cells in response to intracellular MRSA infection showed an activation in the glycolysis pathway [12].

### 2.2. Docosanoic, Eicosanoic and Palmitic Acids Showed a Cytoprotective Role in MRSA-Infected HeLa Cells

We recently observed that the silencing of ELOVL1 during *S. aureus* infection leads to an increase of host cell death [27]. ELOVL1 participates in the production of very long-chain fatty acids synthesis by catalyzing the first reaction of their elongation cycle [28]. Hence, we hypothesized that the production of very long-chain fatty acids may have a cytoprotective role during cell infection. We therefore evaluated the impact of docosanoic, eicosanoic and palmitic acids on intracellular MRSA infection if given exogenously. To this end, we treated HeLa cells with 30 µM of docosanoic, eicosanoic or palmitic acid and measured the host cell viability after MRSA infection. We employed stearic acid as an additional control, since the levels of this fatty acid were not increased in MRSA-infected cells (Figure 3).

When compared to the negative control (DMEM supplemented with DMSO), the host cell viability was significantly increased after MRSA infection when DMEM was supplemented with either docosanoic, eicosanoic or palmitic acids (Figure 3A). However, host cell viability was not restored when HeLa cells were treated with stearic acid (Figure 3A).

We speculated that host cell viability was partially restored in the presence of these saturated fatty acids due to a direct reduction or inhibition of bacterial growth. However, we did not detect any statistically significant changes when the intracellular MRSA survival was quantified across the different conditions (Figure 3B). In contrast, when we measured in vitro bacterial growth in Nutrient Broth medium supplemented with the aforementioned cytoprotective fatty acids we observed that MRSA growth at 6 and 8 h was significantly reduced in the presence of both docosanoic and palmitic acids (Figure 4). At 24 h, MRSA growth was attenuated by all the cytoprotective fatty acids—docosanoic, eicosanoic and palmitic acids—but not by the presence of stearic acid in the medium (Figure 4B).

Overall, these results suggest that the increase of host cell viability after MRSA infection is due to the direct inhibition of MRSA growth in the presence of these three fatty acids, which antimicrobial activity may be bacteriostatic. This would explain why intracellular CFU counts were not altered across conditions, whilst bacterial in vitro growth was attenuated at long periods of incubation.

The antibacterial properties of lipids and fatty acids have been proposed since 1880s, when Robert Koch and his colleagues showed that specific fatty acids could inhibit the growth of *Bacillus anthracis* [29]. From then, many other studies have proven the bactericidal or bacteriostatic effect of several saturated and unsaturated fatty acids on Gram-positive and Gram-negative bacteria [20,30]. Nowadays, partially driven by the emergence of multidrug-resistant strains, the research on fatty acids and monoglycerides as potential antibacterial agents has gained new attention [30,31,32,33,34,35,36].

Fatty acids are amphipathic molecules that can promote membrane destabilization and pore formation, eventually leading to inhibition of bacterial growth or bacterial cell death. Targeting bacterial cell membrane is the key antibacterial effect of fatty acids and the mechanism of action includes: (i) increase in the membrane permeability, and consequently, cell lysis, (ii) disruption of the electron transport chain and uncoupling oxidative phosphorylation, and (iii) inhibition of membrane enzymes and nutrient uptake [30].

Specifically, several studies have shown that specific saturated and unsaturated fatty acids play an antibacterial role against *S. aureus*. As an example, topical administration of lauric acid have shown to be effective in vitro and in vivo to treat skin infections, caused by *S. aureus* and *Propionibacterium acnes* [37,38]. The delivery of antimicrobial lipids in highly concentrated forms—by formulation in liposomes or nanoparticles—has also been tested to treat *Helicobacter pylori* infection in vitro and in vivo [39,40,41]. Medium-chain length fatty acids, e.g., lauric acid are considered one of the most potent saturated fatty acid against Gram-positive bacteria, including Methicillin-susceptible *Staphylococcus aureus* (MSSA) and MRSA strains, albeit at high MOIs [21,42]. Similarly, treatment with arachidonic acid was effective against most Gram-positive bacteria, including *S. aureus* [43]. Furthermore, treatment with the linolenic acid also inhibits the growth of this pathogen [44]. The inhibition of bacterial cell growth by linolenic acid is achieved by increasing membrane permeability, which blocks macromolecular synthesis and disrupts the electron transport chain [45]. Some unsaturated fatty acids—including linolenic acid—have also shown antibacterial activity by inhibiting the enoyl-acyl carrier protein reductase (FabI), which is necessary for the fatty acid elongation process [46].

The three saturated fatty acids identified in this study—docosanoic, eicosanoic and palmitic acids—significantly reduced in vitro MRSA growth and restored host cell viability after infection. Nevertheless, the precise mechanism underlying the antibacterial activity of docosanoic, eicosanoic and palmitic acids remains unknown and further experiments need to be carried out to unravel the underlying mechanistic. It is, for example, currently unknown whether the fatty acids are bound to specific membranes of the host cell, e.g., the phagosome. Interestingly, certain host lipases may protect the infected cell against intracellular pathogenic bacteria by releasing antimicrobial lipids from the host cell membrane [47]. However, the cell response to the free fatty acids should be further characterized, as the observed protective effect on host cell viability may be due to other factors, such as a higher resistance to cellular death.

In summary, antibacterial fatty acids possess a great potential to combat bacterial infections. In addition, the combination of fatty acids along with existing therapies may increase their effect and could contribute to the reduction of antimicrobial resistance. The combination of free fatty acids and cholesteryl esters have been recently tested against *Pseudomonas aeruginosa* and *Staphylococcus epidermidis*, showing potential as novel antimicrobial agents [48].

## 3. Materials and Methods

### 3.1. Bacterial Strains, Cell Lines and Culture Conditions

*S. aureus* USA300 LAC strain [49] was cultured in Nutrient Broth (NB) medium (Sigma-Aldrich, Gillingham, UK). Routinely, bacteria were incubated at 37 °C, with vigorous shaking (300 rpm). For growth on solid medium, Nutrient Agar plates (NA; Sigma-Aldrich) were used and incubated at 37 °C.

To evaluate in vitro bacterial growth, *S. aureus* was incubated in 96-well plates with NB supplemented with either 100 µg/mL gentamycin or 30 µM of docosanoic, eicosanoic, palmitic, or stearic acids; DMSO was used as negative control. Optical density at 600 nm (OD_600 nm_) was measured at different time points using an EL800 Microplate reader (Bio-Tek, Winooski, VT, USA).

For preinocula preparation, *S. aureus* USA300 LAC strain was grown overnight in 10 mL of NB at 37 °C with shaking (300 rpm) and 0.5 mL of this culture was used to inoculate a flask containing 50 mL of NB. Bacterial cultures were grown until OD_600 nm_ of 1 was reached, upon which the culture was centrifuged (4000 rpm, 15 min, 4 °C). Pellets were then washed twice with Dulbecco’s Phosphate-Buffered Saline (DPBS), resuspended in 1.5 mL of PBS supplemented with 20% glycerol and aliquots of 100 µL were stored at −80 °C until needed. Preinocula concentration was calculated by serial dilution plating and colony forming units (CFU) counting.

HeLa cells (ECACC 93021013) were grown in 100 mm cell culture plates (Sarstedt) with Dulbecco’s Modified Eagle’s medium (DMEM) containing pyruvate, glucose and glutamine and supplemented with 10% heat-inactivated foetal bovine serum (FBS) and 5% of penicillin and streptomycin solution, unless otherwise specified. Cells were incubated at 37 °C and 5% of CO_2_ in a Heraeus^®^ BB15 incubator.

### 3.2. Intracellular Infection Assays

Intracellular MRSA infection assays were carried out as previously described [12,50]. For metabolomics approaches, HeLa cells were seeded in 6-well plates in DMEM without antibiotics at a cell density of 5 × 10^5^ cells per well. For host cell viability assays, HeLa cells were seeded in 24-well plates in DMEM without antibiotics at a cell density of 7.5 × 10^4^ cells per well. In both cases, cells were incubated overnight. The next day, an aliquot of bacterial preinocula was thawed, diluted in 900 µL of PBS and centrifuged at 4000 rpm for 5 min. Pellet was washed twice with PBS prior to addition of DMEM without antibiotics at a bacterial density that corresponds to a multiplicity of infection (MOI) of 100. Then, the bacterial suspension supplemented with DMSO or 30 µM of docosanoic, eicosanoic, palmitic, or stearic acids was added to each well, plates were centrifuged immediately at 1500 rpm for 5 min and incubated at 37 °C in 5% CO_2_ for 45 min to allow bacterial internalization. The medium was then replaced by DMEM supplemented with DMSO or 30 µM of the afore mentioned fatty acids and 100 µg/mL gentamycin to kill extracellular bacteria, and the plates were placed back in the incubator until the desired time points were reached.

### 3.3. Host Cell and Bacterial Viability Assays

Host cell viability was quantified after 6 h of infection as previously described [50]. Briefly, to recover both necrotic and apoptotic cells, supernatants of each well were transferred into clean Eppendorf’s tubes. Cells were then trypsinized, diluted in DMEM without antibiotics, mixed with the supernatants and centrifuged at 1500 rpm for 10 min. Afterwards, cells were double stained with annexin V-FITC and propidium iodide according to manufacturer’s recommendations (Becton Dickinson, BD; Wokingham, UK), diluted in 50 µL of DMEM per sample and incubated for 15 min at room temperature. Stained cells were centrifuged at 1500 rpm for 10 min, supernatant was aspirated, and cell pellets were fixed using 150 µL of BD Cytofix buffer (Becton Dickinson, BD; Wokingham, UK) for 15 min at 4 °C. Finally, samples were diluted with 350 µL of DMEM and host cell viability was measured by flow cytometry (BD Accuri™ C6 Plus).

To quantify intracellular MRSA viability, infected HeLa cells were lysed at 6 h post infection with 0.1% Triton X-100 diluted in PBS. The samples were then serially diluted and plated on Nutrient Agar for CFU/mL determination.

### 3.4. Samples Preparation for Intracellular Metabolome Analysis

After 6 h of infection, cells were washed with 1 mL of ice-cold Ringer’s solution and quenched by adding 1 mL of cold (−20 °C) LC-MS grade methanol [51]. Cells were then detached by using a cell scraper and the cold methanol suspension was transferred into a clean Eppendorf tube. Extraction was repeated with a further 0.5 mL of cold methanol and extracts were pooled and stored at −80 °C.

To obtain the organic fraction, extracts were first dried using an Eppendorf Vacufuge Concentrator. A dual phase extraction was performed by adding 300 µL of CHCl_3_/MeOH (2:1) and vortexing for 30 s. After addition of 300 µL of water and centrifugation (13,000 rpm, 10 min, RT), the lower organic layers were placed into glass vials and dried overnight before being stored at −80 °C.

For derivatization, the organic fraction was reconstituted in 300 µL of methanol/toluene solution (1:1 ratio), treated with 200 µL of 0.5 M sodium methoxide and incubated for 1 h at room temperature. Reaction was stopped by adding 500 µL of 1M NaCl and 25 µL of concentrated HCl. Fatty acids were extracted by using 500 µL of hexane and organic layers were dried in the fume cupboard under N_2_. Organic samples were then derivatized with 40 µL acetonitrile and 40 µL of N-(tert-butyldimethylsilyl)-N-methyltrifluro-acetamide (MBTSFA; Thermo Fisher Scientific, Waltham, MA, USA) and incubated at 70 °C for 1 h. Samples were finally centrifuged at 2000 rpm for 5 min prior to transferring them into a clean vial for GC-MS analysis.

### 3.5. Gas Chromatography-Mass Spectrometry (GC-MS)

GC-MS analysis was performed as previously described [12]. Briefly, analysis was carried out on an Agilent 7890 GC equipped with a 30 m DB-5MS capillary column with a 10 m Duraguard column connected to an Agilent 5975 MSD operating under electron impact (EI) ionization (Agilent Technologies, Santa Clara, CA, USA). Samples were injected with an Agilent 7693 autosampler injector into deactivated spitless liners using helium as the carrier gas [24].

Metabolites were identified and quantified using a workflow described previously [52]. Samples were deconvoluted in AMDIS [53,54] and quantified using an in-house script. Integration of labelled metabolites was carried out based on an in-house fragment/retention time database using an updated version of the Matlab script capable of natural isotope correction [24,55].

### 3.6. Statistical Analysis

Statistical analysis was conducted using GraphPad Prism software. One-way ANOVA and post hoc Tukey’s multiple comparison tests were employed to examine significant differences across treatments.

## 4. Conclusions

The host-lipidome is altered in response to intracellular MRSA infection. We found that the absolute levels of docosanoic, eicosanoic, and palmitic acids are significantly increased in MRSA-infected cells. Interestingly, we observed that host cell viability is partially restored after MRSA infection in the presence of these three fatty acids, while in vitro bacterial growth is hampered. Therefore, docosanoic, eicosanoic and stearic acids seems to play a cytoprotective role in HeLa cells due to direct reduction of MRSA in vitro growth. Further research is needed to fully elucidate the antibacterial mechanism of action of these three fatty acids.

## Figures and Tables

**Figure 1 metabolites-09-00148-f001:**
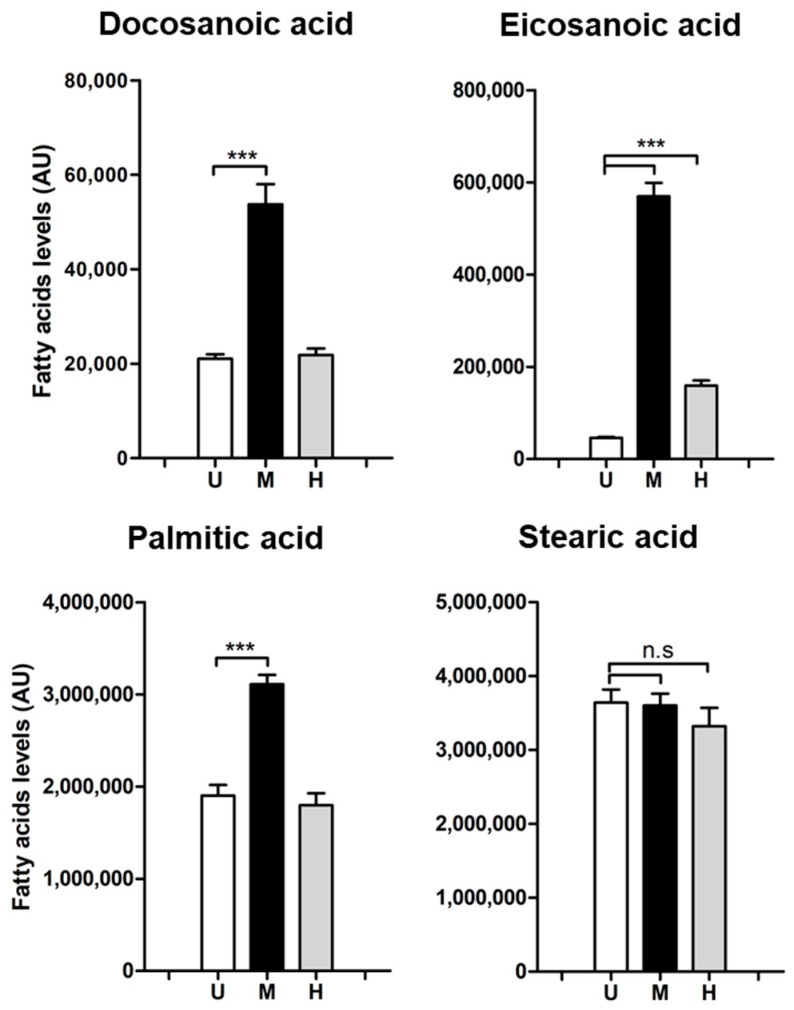
Absolute levels of lipid-bound docosanoic, eicosanoic and palmitic acid are affected after intracellular USA300 infection. HeLa cells were infected with USA300 and exposed to heat-killed USA300 (MOI 100; 6 h) and fatty acids levels were detected by GC-MS. Bar graphs show absolute levels of each fatty acid in uninfected cells (white bars), cells infected with USA300 strain (black bars) and cells exposed to heat-killed USA300 (grey bars). Data are expressed as means ± standard error (SE) of three independent experiments performed in triplicates. One-way ANOVA and post hoc Tukey’s multiple comparison tests were performed to assess statistically significance across treatments. *p*-value ≤ 0.001 (***); n.s (no significant differences). U = Uninfected cells; M = methicillin-resistant *Staphylococcus aureus* (MRSA)-infected cells; H = cells exposed to heat-killed bacteria.

**Figure 2 metabolites-09-00148-f002:**
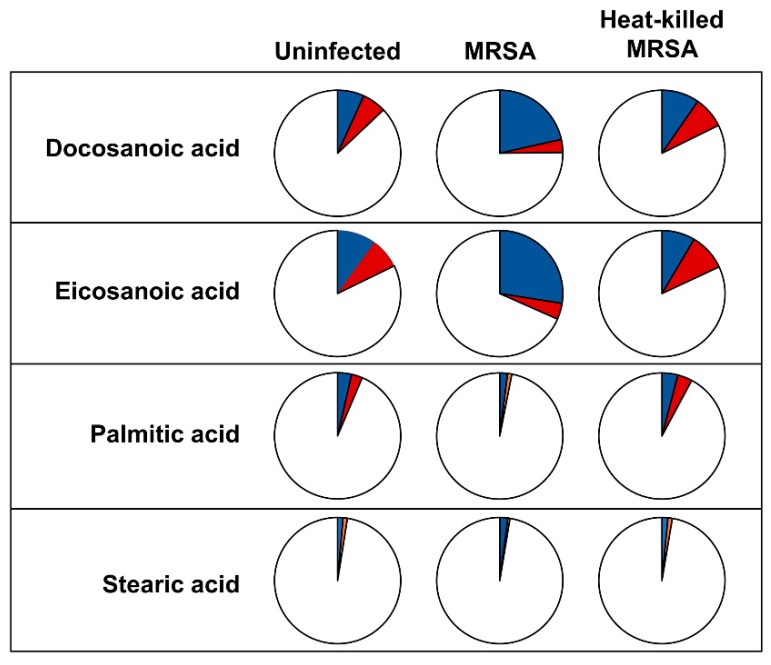
Labelling distribution of lipid-bound docosanoic, eicosanoic and palmitic acid is altered after intracellular USA300 infection. HeLa cells were infected with USA300 and exposed to heat-killed USA300 (MOI 100; 6 h) and labelling distribution was detected by GC-MS. Pie charts display the labelling pattern of transesterified fatty acids from uninfected cells (left pie-chart), cells infected with USA300 strain (middle pie-chart) and cells exposed to heat-killed USA300 (right pie-chart). Within pie-charts, blue slides represent carbon coming from labelled glucose, whereas red slides are carbon coming from labelled glutamine and white slides show carbon from other sources.

**Figure 3 metabolites-09-00148-f003:**
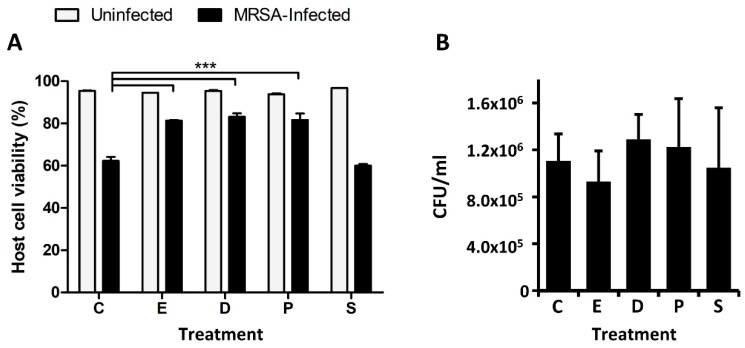
Treating HeLa cells with eicosanoic, docosanoic or palmitic acid increases host cell viability after USA300 infection but does not affect MRSA intracellular survival. HeLa cells were infected with USA300 (MOI 100; 6 h) in the presence of DMSO (**C**), docosanoic acid (**D**), eicosanoic acid (**E**), palmitic acid (**P**) and stearic acid (**S**), and host cell viability was quantified. (**A**) Quantification of host cell viability after USA300 infection was measured by flow cytometry using a double annexin V-FITC and PI staining. (**B**) Intracellular MRSA survival quantified by colony forming units per milliliter (CFU/mL). Data are expressed as means ± SE of three independent experiments performed in duplicates. One-way ANOVA and post hoc Tukey’s multiple comparison tests were performed to assess statistically significance across treatments. *p*-value ≤ 0.001 (***).

**Figure 4 metabolites-09-00148-f004:**
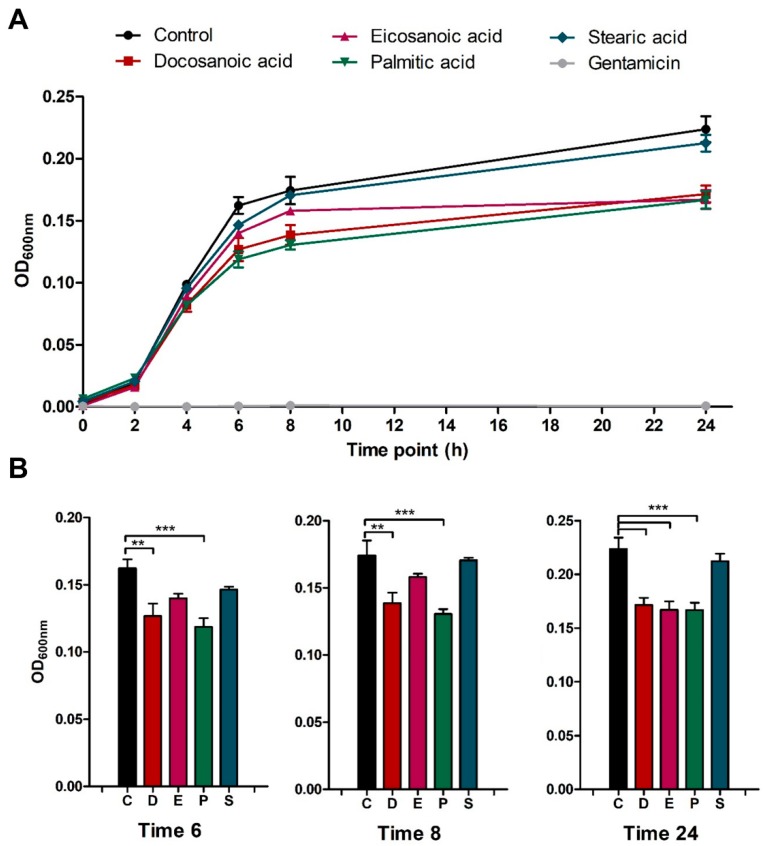
Eicosanoic, docosanoic and palmitic acid restricts in vitro USA300 growth in Nutrient Broth medium. (**A**) In vitro bacterial growth curves in Nutrient Broth and supplemented with DMSO (control), one fatty acid (30 µM) and gentamicin (100 µg/mL) were evaluated by measuring the absorbance at 2, 4, 6, 8 and 24 h. (**B**) In vitro bacterial growth at 6, 8 and 24 h in the presence of DMSO (**C**), docosanoic acid (**D**), eicosanoic acid (**E**), palmitic acid (**P**) or stearic acid (**S**). Data are expressed as means ± SE of three independent experiments performed in duplicates. One-way ANOVA and post hoc Tukey’s multiple comparison tests were performed to assess statistically significance across treatments. *p*-value ≤ 0.01 (**); ≤ 0.001 (***).

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
