# Peer review of "Intracellular Staphylococcus aureus Elicits the Production of Host Very Long-Chain Saturated Fatty Acids with Antimicrobial Activity"

_metabolites, 2019, doi:10.3390/metabo9070148_

Reviewer 1 Report

In the manuscript titled “Intracellular Staphylococcus aureus elicits the production of host very long-chain fatty acids with antimicrobial activity”, Bravo-Santana et al. investigate the role that host fatty acids play during intracellular infection with S. aureus. The authors utilize an in vitro model of infection, infecting HeLa cells with the methicillin-resistant S. aureus (MRSA) strain USA300. Using mass spectrometry, the authors identify changes in lipid-bound fatty acids when infected with MRSA. They highlight increased abundances of lipids containing docosanoic acid, eicosanoic acid, and palmitic acid upon S. aureus infection. To identify synthesis of lipids, they feed HeLa cells radiolabeled glucose (for glycolytic flux) and glutamine (for carbon sources other than glucose) and infect the cells with MRSA. They identify that the increase in production of the above fatty acids is driven by glycolysis. Finally, they show that pre-treatment of HeLa cells with docosanoic, eicosanoic, and palmitic acid protect host cells from intracellular MRSA infection due to the antibacterial properties of these fatty acids. In conclusion, they show that host cells modulate lipid-bound fatty acids upon exposure to MRSA to help prevent infection in vitro. This is an interesting paper that identifies new host responses to S. aureus. The idea is solid and I think the techniques produce data that support their hypothesis. However, I’m not sure how to interpret their wording of “lipid-bound fatty acids” and how they are effective against MRSA compared to their unbound free fatty acid counterparts (see below).

Major comments:

1. The authors mention that ELOVL1 is important for synthesizing these long-chain fatty acids. What would happen if you inhibit/knock-down this gene in HeLa cells then pre-treat the cells with the indicated fatty acids?

2. The authors measure “lipid-bound fatty acids” after MRSA infection using mass spectrometry. However, when they treat host cells as well as determine sensitivity of MRSA to the fatty acids, they use unbound free fatty acids. This drives their conclusion that lipids containing these fatty acids help protect host cells. When measuring fatty acids by mass spectrometry, can you identify whether a fatty acid is free or bound to glycerol as a lipid? Are these lipid-bound fatty acids still bioavailable to inhibit S. aureus growth? When measuring the changes in fatty acids of the host cells upon MRSA infection, are these lipid-bound fatty acids liberated to become available to inhibit S. aureus? It is not clear how lipid-bound fatty acids would inhibit MRSA directly. It is entirely possible that free fatty acids pools resemble the lipid-bound fatty acids, but this was not tested.

3. It would be interesting to test whether the pretreatment of HELA cells with fatty acids, followed by washing the cells before MRSA addition alters the response. This would demonstrate whether the fatty acids elicit a cellular response that is inhibitory to MRSA that is not the direct activity of the fatty acids on MRSA.

Minor comments

Figure 4. Why does Sa only grow to an OD600 of less than 0.25 in these assays. This seems like very poor growth. Indeed, that methods indicate that MRSA grows to much higher OD600 in a flask. The OD reported seems quite low to many other studies of MRSA grown in plate readers.

Figure 4, Results. The sensitivity of MRSA to fatty acids has been well documented. There are several recent papers that should be referenced for this in addition to the much older literature used here.

Line 63. Ref 17 has nothing to do with MRSA.

Line 66. Sa can insert exogenous fatty acids into the membrane, but cannot use fatty acids as a carbon source. This has been demonstrated.

Author Response

Major comments:

1. The authors mention that ELOVL1 is important for synthesizing these long-chain fatty acids. What would happen if you inhibit/knock-down this gene in HeLa cells then pre-treat the cells with the indicated fatty acids?

As mentioned in the manuscript, we have recently observed that the silencing of ELOVL1 during S. aureus infection leads to an increase of host cell death. These results are part of a genome-wide unbiased shRNA screening performed to find novel targets for host-directed therapeutics against intracellular MRSA. The results were presented in an oral presentation at the International Symposium on Staphylococci and Staphylococcal infections, University of Copenhagen, Denmark.; 2018, and the data will be published as part of a manuscript that is currently under revision in Scientific Reports (Ref. SREP-19-21231).

2. The authors measure “lipid-bound fatty acids” after MRSA infection using mass spectrometry. However, when they treat host cells as well as determine sensitivity of MRSA to the fatty acids, they use unbound free fatty acids. This drives their conclusion that lipids containing these fatty acids help protect host cells. When measuring fatty acids by mass spectrometry, can you identify whether a fatty acid is free or bound to glycerol as a lipid? Are these lipid-bound fatty acids still bioavailable to inhibit S. aureus growth? When measuring the changes in fatty acids of the host cells upon MRSA infection, are these lipid-bound fatty acids liberated to become available to inhibit S. aureus? It is not clear how lipid-bound fatty acids would inhibit MRSA directly. It is entirely possible that free fatty acids pools resemble the lipid-bound fatty acids, but this was not tested.

The lipid-bound fatty acids are transesterified and converted to fatty acid methyl esters (FAMEs) during derivatisation. This derivatisation does not affect free fatty acids. On the other hand, we can only speculate that certain lipases may release those lipid-bound fatty acids during infection. It has been previously suggested that certain host lipases may have a role in host defence against pathogenic bacteria by releasing arachidonic acid from the host cell membrane (J Adv Res. 2018 Jan 3;11:57-66. doi: 10.1016/j.jare.2018.01.001). This is now discussed in the manuscript (Lines 201-203).

3. It would be interesting to test whether the pretreatment of HELA cells with fatty acids, followed by washing the cells before MRSA addition alters the response. This would demonstrate whether the fatty acids elicit a cellular response that is inhibitory to MRSA that is not the direct activity of the fatty acids on MRSA.

Many thanks for the suggestion. We have considered to study in detail the cell response to these lipids. Unfortunately, we do not have currently the resources to perform any further experiments.

Minor comments

Figure 4. Why does Sa only grow to an OD600 of less than 0.25 in these assays. This seems like very poor growth. Indeed, that methods indicate that MRSA grows to much higher OD600 in a flask. The OD reported seems quite low to many other studies of MRSA grown in plate readers.

The quantification of bacterial in vitro growth was performed in 96 wells-plates with a maximum volume of 100ul, which leads to lower ODs readouts as compared to when bacterial growth is measured in microcuvettes and a traditional spectrophotometer.

Figure 4, Results. The sensitivity of MRSA to fatty acids has been well documented. There are several recent papers that should be referenced for this in addition to the much older literature used here.

Many thanks for the suggestion, we have added to the bibliography some more recent papers on this topic (Lines 174-175).

Line 63. Ref 17 has nothing to do with MRSA.

The in-text citation was misplaced in that sentence, this is now amended (Line 62).

Line 66. Sa can insert exogenous fatty acids into the membrane, but cannot use fatty acids as a carbon source. This has been demonstrated.

The sentence has been amended accordingly (Lines 65-66). 

Reviewer 2 Report

The manuscript by Bravo-Santano et al. describes the cellular increase of 3 fatty acids upon infection with MRSA, fatty acids that appear to enhance cell survival upon infection and have a potential antimicrobial effect. It has been previously shown that fatty acids are important in MRSA infection but it is not known whether they are playing a role as immune modulators or if they are important carbon sources for the pathogen. 

Here the authors show that docosanoic, eicosanoic and palmitic acid levels are increased in HeLa cells infected with MRSA and that most of the elongation of the fatty acids is driven from glucose, data that fits with their previous report of activation of the glycolysis pathway on HeLa cells infected with MRSA. Treatment of HeLa cells infected with MRSA with the different fatty acids seems to enhance their survival, suggesting that these might be important as antimicrobials. Additionally, the three fatty acids also seem to slightly reduce the growth of MRSA in liquid culture, pointing once again into a potential antimicrobial effect. 

The manuscript is well written and the figures are well presented and adequate for the results described in the text. This short communication also brings interesting insights into the role played by these fatty acids during MRSA infection and lays the foundation for further studies on the mechanism of action of these fatty acids as potential antimicrobial molecules. 

Comments: 

1) How was infection evaluated? The manuscript describes how HeLa cells were infected but infection is not ''quantified'' or demonstrated anywhere. 

2) Line 23: The authors state "In addition, treatment of HeLa cells with these three fatty acids showed a cytoprotective role by directly reducing S. aureus growth." but this is not shown in the manuscript. 

Although they have shown that HeLa cells treatment with the three fatty acids increases cell viability and that the three fatty acids seem to moderately affect the growth of MRSA USA 300 in vitro (broth culture), it is not shown that increased cell viability is due to reduced numbers of intracellular MRSA  and the authors have not shown that any of the fatty acids has an effect on intracellular bacteria, only on bacteria grown on liquid media. Effects of different molecules on intracellular bacteria will depend on diverse factors and this was not evaluated/demonstrated in this work. 

3) Lines 122-162: The authors show in their results and discussion section that they have treated cells with the different fatty acids and also grown MRSA in broth medium supplemented with the different fatty acids, but do not describe in their methods section how any of these experiments were performed, for e.g., how long where cells treated, where they pre-treated or incubated with the different fatty acids during/before/after infection, etc. 

4) Line 128: The authors have used 30 μM of the different fatty acids to treat cells but do not explain why they have chosen this concentration for treatment (decided after titration with different concentrations, how this concentration compares to physiological concentrations, etc). 

5) Lines 145-151: It is not stated the concentration of each of the fatty acids used for the MRSA growth inhibition assays and how where these performed, both here and in the methods section. 

6) Line 152 and Line 283: I think this should be reformulated as several other factors might be playing a role in the observed effect and the effect of these fatty acids on intracellular bacteria was not demonstrated. 

7) The methods section is missing several details that are important for the correct understanding of the manuscript and for the correct evaluation of the results (e.g. previously mentioned on comments 2 and 4).

Author Response

Comments:

1) How was infection evaluated? The manuscript describes how HeLa cells were infected but infection is not ''quantified'' or demonstrated anywhere.

Many thanks for this comment, we have quantified MRSA intracellular survival by CFU counting, the results are now included in Figure 3B.

2) Line 23: The authors state "In addition, treatment of HeLa cells with these three fatty acids showed a cytoprotective role by directly reducing S. aureus growth." but this is not shown in the manuscript.

Although they have shown that HeLa cells treatment with the three fatty acids increases cell viability and that the three fatty acids seem to moderately affect the growth of MRSA USA 300 in vitro (broth culture), it is not shown that increased cell viability is due to reduced numbers of intracellular MRSA  and the authors have not shown that any of the fatty acids has an effect on intracellular bacteria, only on bacteria grown on liquid media. Effects of different molecules on intracellular bacteria will depend on diverse factors and this was not evaluated/demonstrated in this work.

The referee is right, we have now quantified intracellular MRSA and the treatment does not lead to reduced bacterial viability (Figure 3B), suggesting that the three lipids have bacteriostatic activity instead of bactericidal, which fits with the moderate effect on the in vitro growth of USA300 at very late time points. This is now mentioned in the manuscript (Lines 147-159).

3) Lines 122-162: The authors show in their results and discussion section that they have treated cells with the different fatty acids and also grown MRSA in broth medium supplemented with the different fatty acids, but do not describe in their methods section how any of these experiments were performed, for e.g., how long where cells treated, where they pre-treated or incubated with the different fatty acids during/before/after infection, etc.

Many thanks for pointing out this omission. The requested information is now included in Materials and Methods (Lines 241-246).

4) Line 128: The authors have used 30 μM of the different fatty acids to treat cells but do not explain why they have chosen this concentration for treatment (decided after titration with different concentrations, how this concentration compares to physiological concentrations, etc).

30 μM was the maximum soluble concentration that we could obtain.

5) Lines 145-151: It is not stated the concentration of each of the fatty acids used for the MRSA growth inhibition assays and how where these performed, both here and in the methods section.

This information is now included in Figure 4 and Materials and Methods (Lines 217-219).

6) Line 152 and Line 283: I think this should be reformulated as several other factors might be playing a role in the observed effect and the effect of these fatty acids on intracellular bacteria was not demonstrated.

Yes, we have now discussed this point in lines 201-205.

7) The methods section is missing several details that are important for the correct understanding of the manuscript and for the correct evaluation of the results (e.g. previously mentioned on comments 2 and 4).

Thanks, the materials and methods have been amended accordingly.

Reviewer 3 Report

Well written paper. 

Introduction needs revision. Paragraph 1 and 2 could be combined to avoid repetition and more information on the utilization of host derived molecules by S. aureus and manipulation of host cell metabolism by intracellular bacterial pathogens could provide adequate introduction.

Unsaturated fatty acids and their potential for toxicity to bacteria is well described in the literature. However, the papers that did investigate these phenomena described the toxicity in terms of loss of membrane potential etc. Lower OD600 values that the authors demonstrate could merely be an indication of a bacteriostatic effect. Suggest the authors perform S. aureus viability assays in the presence of these fatty acids (DEP) and demonstrate toxicity of these fatty acids to S. aureus.

There is evidence that S. aureus membrane fatty acid composition is plastic and can incorporate host derived fatty acids into its membrane either with or without further modification (See- Sen et al 2016 10.1371/journal.pone.0165300). Suggest the authors discuss the implications of alterations in the host cell fatty acid composition in terms of the intracellular bacterium that they are projected to contain.

Author Response

Introduction needs revision. Paragraph 1 and 2 could be combined to avoid repetition and more information on the utilization of host derived molecules by S. aureus and manipulation of host cell metabolism by intracellular bacterial pathogens could provide adequate introduction.

The introduction has been amended following these suggestions (Lines 33-36 and 41-45).

Unsaturated fatty acids and their potential for toxicity to bacteria is well described in the literature. However, the papers that did investigate these phenomena described the toxicity in terms of loss of membrane potential etc. Lower OD600 values that the authors demonstrate could merely be an indication of a bacteriostatic effect. Suggest the authors perform S. aureus viability assays in the presence of these fatty acids (DEP) and demonstrate toxicity of these fatty acids to S. aureus.

Many thanks for this comment, we have evaluated intracellular MRSA viability (Fig 3B) and the results suggest indeed that the free fatty acids have a bacteriostatic effect. This is now also mentioned in the results and discussion (Lines 147-159).

There is evidence that S. aureus membrane fatty acid composition is plastic and can incorporate host derived fatty acids into its membrane either with or without further modification (See- Sen et al 2016 10.1371/journal.pone.0165300). Suggest the authors discuss the implications of alterations in the host cell fatty acid composition in terms of the intracellular bacterium that they are projected to contain.

This is now mentioned in the introduction (Lines 65-67).

Round  2

Reviewer 1 Report

My only final comment is about the referencing of new literature for sensitivity of S. aureus to fatty acids. The authors state that they added more (lines 174-175). My amended draft does not have any line numbers for 171-181. I assume they are referring to the original line numbers but I don't see any differences here.